# Dynamics Optimization Research and Dynamics Accuracy and Reliability Analysis of a Multi-Link Mechanism with Clearances

**Shuai Jiang** [1]**, Yuanpeng Lin** [2]**, Jianan Liu** [2]**, Linjing Xiao** [1] **and Shuaishuai Zhang** [1,*]

[1] Department of Electrical and Information, Shandong University of Science and Technology, Jinan 250031, China
[2] Swinburne College, Shandong University of Science and Technology, Jinan 250031, China
**\*** Correspondence: zhangshuais@sdust.edu.cn; Tel.: +86-15253171034

**Abstract:** With the development of high-speed and lightweight mechanisms, and the continuous improvement of manufacturing accuracy requirements in industrial production, clearance joints have increasingly become one of the key factors affecting dynamics performance. Poor clearance will seriously compromise stability, accuracy, and dynamics performance. Based on a genetic algorithm, an efficient modeling methodology for the dynamics optimization of a planar complex multi-link mechanism containing multiple clearance joints is put forward. The model comprises a 2-degree of freedom (*DOF*) nine-bar mechanism that can be used as the main transmission mechanism of a hybrid drive multi-link press, which is taken as the research object. The optimization objective is to minimize the maximum acceleration of the slider and minimize the difference between the actual central trajectory and the ideal trajectory. By optimizing the quality parameters of key components, an optimal solution for the design parameters is obtained, and the effects of the different optimizations of the objective functions on dynamics response are compared and analyzed. At the same time, a new modeling and calculation methodology of the dynamics accuracy and reliability of a complex multi-link mechanism in terms of multiple clearances is proposed, and the effect of optimization on dynamics accuracy and the reliability of the mechanism is analyzed. Based on the optimization results obtained by taking the minimum difference between the actual center trajectory and the ideal trajectory as an optimization objective, the nonlinear characteristics before and after optimization are analyzed through a phase diagram and Poincaré map. A test platform was built to study the dynamics of the mechanism with clearances. Research not only provides a basis for the dynamics optimization of a multi-link mechanism containing clearances but also provides reference significance for the reliability analysis of a multi-link mechanism containing clearances.

**Keywords:** dynamics optimization; clearances; dynamics response; dynamics accuracy reliability; nonlinear characteristics





## 1. Introduction

As an important element of mechanical structures, the multi-link mechanism has the advantages of a simple structure, good reliability, and strong bearing capacity. In recent years, it has been the focus of mechanism research and has attracted the attention of many scholars. In the analysis of such a mechanism's motion, it is usually assumed that a kinematic pairing is ideal. However, in practice, the clearance of kinematic pairing is inevitable due to tolerances in the design, manufacturing defects, and wear. Clearances will increase certain additional and uncontrollable degrees of freedom, cause additional wear, and impact various elements in the kinematic pair. It will also lead to the deterioration of dynamics performance or adverse vibrations and will mean that the dynamics behavior of the actual mechanism with clearance will deviate from the dynamics behavior of an ideal mechanism. The degree of freedom brought by a clearance joint is a significant

source of error [1–11]. Aiming at the problems of low stability and intensified wear during the operation of a mechanism containing clearances, according to its kinematic and dynamics characteristics, we addressed this problem through the dynamics optimization of a mechanism with a clearance. The vibration and noise caused by poor clearance can be effectively reduced and the stability and reliability of the mechanism could be improved [12–14].

How to reduce the adverse effects of clearances on dynamics performance and improve the accuracy and performance of mechanisms is a hot topic. Many scholars have carried out a series of theoretical studies examining this problem. Based on the continuous contact model, some scholars have carried out the dynamics optimization of a four-bar mechanism containing revolute clearances [15,16]. Sardashti et al. [15] assumed a four-bar mechanism, considering a single clearance as an example, and proposed an algorithm based upon particle swarm optimization to solve the optimization problem. Erkaya et al. [16] regarded kinematic clearance as a massless virtual rigid bar; they established a dynamics equation for a four-bar mechanism, considering revolute clearance using the Lagrange equation, and optimized the parameters at joint clearance with a genetic algorithm. Based on the Taguchi method, Meng et al. [17] considered clearance, the pin radius, and the friction factor as controllable factors and used collision force as the noise factor to optimize the transmission mechanism of a high-voltage circuit breaker and found that the three controllable factors can effectively affect the dynamics performance of the mechanism. Bai et al. [18] established the normal contact force at clearance by using a nonlinear spring-damper model. The influence of the clearance of a kinematic pair on the dynamics response of a satellite antenna mechanism is studied, and a dynamics optimization method for a satellite antenna driving mechanism with clearance was proposed. Li et al. [19] analyzed the dynamics response and optimized design of a planar rigid–flexible-coupling crank-slider mechanism with double clearances, established a motion differential equation based on the absolute node coordinate method, and studied the effects of a single clearance joint, double clearance joints, and harmonic drive on rigid body dynamics and the rigid–flexible coupling dynamics of the mechanism. Varedi et al. [20] proposed an optimization methodology for a crank-slider mechanism containing clearances based on the particle swarm optimization algorithm, which reduces or eliminates the collision force at the clearance joint by optimizing the mass, centroid position, the moment of inertia of the connection rod, and the mass of the end effector. Sun et al. [21] took a crank-slider mechanism as an example to carry out kinematic analysis and a robust optimization design for a mechanical system containing clearances and proposed a prediction methodology, based on the Baumgarte method and confidence region methodology, to analyze the motion error of a mechanical system. In order to improve the pointing accuracy of a satellite antenna, Ding et al. [22] took the deployment mechanism of a plate satellite antenna as the research object, studied the influence of revolute clearance on pointing accuracy, proposed an analytical method of block modeling using the matrix method, optimized the model based on the particle swarm optimization algorithm, and obtained an optimal solution with a specific configuration. Li et al. [23] optimized the dynamics of the actuator of a multi-link articulated high-voltage circuit breaker with clearance, using the selected clearance size as a design variable. Daniali et al. [24] proposed a comprehensive optimization methodology for the kinematic optimization and dynamics optimization of a four-bar mechanism while considering clearance at the same time, which can reduce the collision of a clearance pair in terms of the dynamics response of the whole mechanism by modifying the mass distribution of the rod, and solved this highly nonlinear optimization problem based upon particle swarm optimization. Ahmedalbashir et al. [25] added a spring between the connecting rod and the swing rod to improve the dynamics responses of a planar four-bar mechanism containing clearances and improve the mechanism's dynamics performance.

Dynamics accuracy refers to the accuracy changes of a mechanism under the influence of working conditions. The reliable performance of dynamics accuracy could better reflect the actual situation of the mechanism. However, research on the dynamics reliability of a

mechanism with clearance has mainly focused on a simple mechanism containing a single clearance, while studies on the reliability of the kinematic and mechanical characteristics of a complex mechanism with multiple clearances were relatively few. Gao et al. [26] analyzed the reliability and sensitivity of a crank-slider mechanism considering multiple clearances and optimized the design of a mechanism containing several clearances, based on an analysis of the design's reliability and sensitivity. Zhang et al. [27] proposed a novel time-dependent reliability methodology to forecast the probability of meeting specific motion requirements within a predetermined time. Through the analysis of many kinds of four-bar mechanisms, the effectiveness of this method was proved. Wei et al. [28] introduced a reliability sensitivity analysis of time-varying parameters and a global reliability sensitivity analysis. An envelope function methodology and the first-order approximation of a motion error function were introduced to effectively estimate several time-dependent PRS and GRS indices. The importance and effectiveness of the proposed methodology were proved with a four-bar mechanism and automobile rack and pinion steering linkage. Zhang et al. [29] took a multi-loop Hoberman radial linkage as their research object, its reliability having been analyzed. Based on the effective length model, the influence of universal joint clearance on dynamics response was established, and the position deviation of the mechanism was analyzed with an improved loop increment methodology. A reliability evaluation model based on the probability density function was established, and the structure of the mechanism was optimized. Chen et al. [30] took a 2-*DOF* seven-bar mechanism containing clearances as an example, for which a dynamics accuracy and reliability model was established based upon the stress strength interference theory. The influences of the different parameters on dynamics accuracy and the reliability of the mechanism were studied. Yu et al. [31] built a comprehensive reliability analysis model of a rolling bearing. By modeling the probability distribution of an actual bearing's working clearance and studying the life factor, the reliability of the results was analyzed. Zhao et al. [32] analyzed the reliability of a slider's displacement of a crank-slider mechanism containing clearance and friction and investigated the effect of different random parameters on the reliability of the slider's displacement.

To sum up, the working of a clearance pair leads to a series of problems, such as poor stability, high noise levels, and intensified wear of the mechanism, which reduces the stability, accuracy, and service life of the mechanism. Therefore, it is urgent to optimize the dynamics of a mechanism with clearance to reduce the negative impact of clearance on the dynamics response of the mechanism. However, the published research on the optimization of a mechanism with clearance mainly focuses on a simple mechanism with a single rotating-pair clearance or on improving the performance of the mechanism by adding springs. There are relatively few studies on the dynamics optimization of a complex mechanism with multiple clearances. The center track at a clearance joint reflects the real motion track of the shaft in the bearing, including their function in free flight mode, continuous contact mode, and impact mode, which directly determines the mechanism's dynamics performance. It is necessary to optimize and improve the dynamics performance of a multi-link mechanism with multiple clearances by reducing the difference between an actual center track and an ideal track, so as to reduce the adverse impact caused by clearance. Clearance exists in almost all motion mechanisms and has a significant impact on motion accuracy. Reliability analysis of a mechanism with clearance has important research significance and application value as a way to improve a mechanism's reliability.

The main aim of this paper is to propose an efficient dynamics optimization method for a planar complex multi-link mechanism including multiple clearances. In this paper, a 2-*DOF* nine-bar mechanism that can be used as the main transmission mechanism of a hybrid drive multi-link press is taken as the research object. In order to improve the dynamics performance of the mechanism, based on the principle of mass distribution, a dynamics optimization model of a 2-*DOF* nine-bar mechanism, considering multiple clearances by using a genetic algorithm is proposed. Two optimization criteria are used to minimize the maximum acceleration of the slider and to minimize the difference between

the actual central trajectory and the ideal trajectory. Based on the optimization results, the nonlinear characteristics before and after optimization are analyzed via a phase diagram and a Poincaré map. The effect of dynamics optimization on the dynamics accuracy and reliability of the mechanism are also analyzed.

The main structure of this paper is as follows. The clearance model is built in Section 2. The dynamics optimization model of the mechanism, containing clearances, is built in Section 3. The dynamics accuracy and reliability model of the mechanism, including the clearances, is established in Section 4. In Section 5, the effects of optimization on the dynamics responses and dynamics accuracy and reliability of a mechanism with clearances are analyzed. A test platform was built to study the dynamics of the mechanism when incorporating clearances. Our conclusions are presented in Section 6.

In view of the adverse effect of the clearance of the kinematic pair on the dynamics response of the mechanism, based on the genetic algorithm, this paper proposes two different dynamics optimization modeling methods for a multi-link mechanism with multiple clearances. Two optimization criteria are used to reduce the influence of clearance on the mechanism, which are to minimize the maximum acceleration of the slider as the optimization objective function and to minimize the difference between the actual center trajectory and the ideal trajectory as the optimization's objective function. According to the optimization results, it was found that when the optimization's objective function is to minimize the difference between the actual trajectory and ideal trajectory, the optimization effect is stronger than that when the optimization objective function is to minimize the maximum acceleration of the slider. The results show that the peak and vibration frequencies of dynamics response are significantly reduced. Based on the optimization results obtained by taking the minimum difference between the actual center trajectory and the ideal trajectory as the optimization objective, the nonlinear characteristics and dynamics accuracy and reliability before and after optimization were analyzed. The results show that optimization improves the nonlinear characteristics and reliability of the mechanism and also makes the mechanism more stable.

## 2. Establishment of the Clearance Model

A clearance model of a revolute pair is shown in Figure 1. $R_1$ and $R_2$ are the radii of the bearing and shaft, respectively. The eccentricity vector of dry friction clearance can be written as:

$$e = r_2^P - r_1^P \tag{1}$$

where $r_1^P$ and $r_2^P$ are the position vectors of the centroid of the shaft and the bearing of the dry friction revolute clearance joint in the fixed coordinate system, respectively.

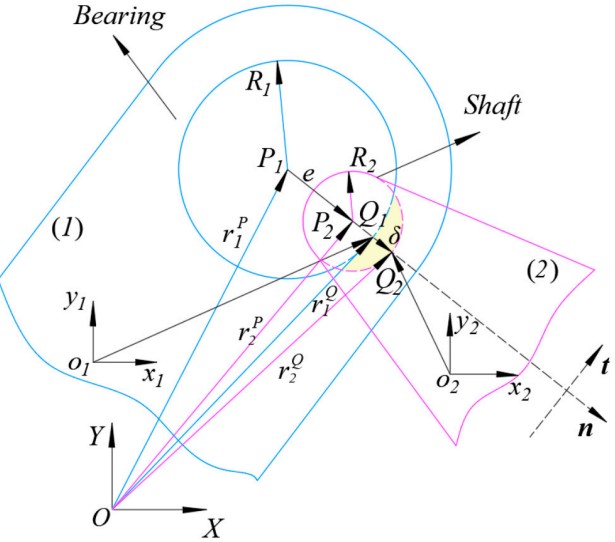

**Figure 1.** Clearance model of a revolute pair.

The unit vector of the eccentric vector is:

$$n = \frac{e}{e}.\tag{2}$$

The embedding depth of clearance of the rotating pair can be expressed as:

$$\delta = e - c \tag{3}$$

where $c$ is the clearance value, $c = R_1 - R_2$, $e$ represents the magnitude of eccentricity vector of dry friction revolute clearance joint, and $e = \sqrt{e \cdot e}$.

Due to the clearance of the revolute pair, according to the geometric relationship between the bearing and shaft, motion states between the shaft and bearing are divided into the free flight state, continuous contact state, and impact state, as shown in Figure 2.

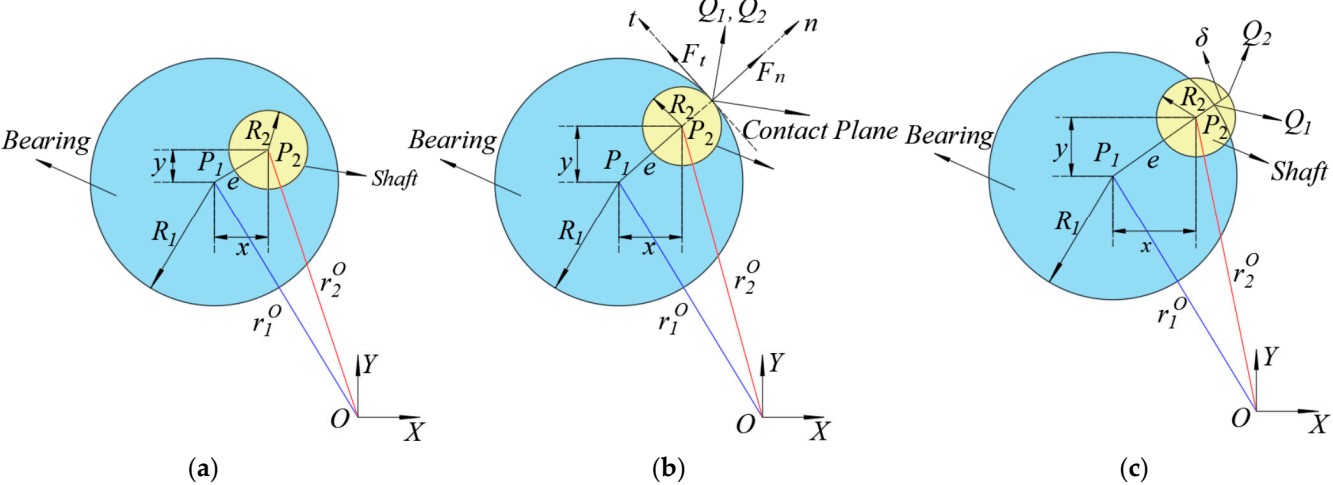

(a)　　　　　　　　　(b)　　　　　　　　　(c)

**Figure 2.** The motion mode of revolute clearance. (**a**) Free flight state; (**b**) continuous contact state; (**c**) impact state.

The criteria for collision between elements in the revolute clearance are as follows:

$$\begin{cases} \delta < 0, \ free flight state \\ \delta = 0, \ continuous contact state \\ \delta > 0, \ impact state \end{cases}. \tag{4}$$

When the embedding depth $\delta < 0$, there is no collision between the shaft and bearing, and they are in free flight mode, with a contact force $F_n = 0$. When the embedding depth $\delta > 0$, the shaft and bearing collide with each other and they are in impact mode, so that the contact force $F_n \neq 0$. When the embedding depth $\delta = 0$, the shaft and bearing are in continuous contact mode, with a contact force $F_n \neq 0$. The change in contact status could be further detected as:

$$\delta(t_n)\delta(t_{n+1}) \leq 0. \tag{5}$$

When $\delta(t_n)\delta(t_{n+1}) \leq 0$, $\delta(t_n) < 0$ and $\delta(t_{n+1}) > 0$, at this moment, the motion state at the clearance joint changes from a free flight state to an impact state. When $\delta(t_n)\delta(t_{n+1}) \leq 0$, $\delta(t_n) > 0$ and $\delta(t_{n+1}) < 0$, at this moment, the motion state at the clearance joint changes from an impact state to a free flight state.

The position vector of the collision point can be expressed as:

$$r_j^Q = r_j^P + R_j n \ (j = 1, 2). \tag{6}$$

When the bearing collides with the shaft, the speed at the contact point is:

$$\dot{r}_j^Q = \dot{r}_j^P + R_j \dot{n} \ (j = 1, 2) \tag{7}$$

where $\dot{n} = \frac{\dot{e}e - \dot{e}e}{e^2}$.

The Lankarani–Nikravesh model (L–N model) is a nonlinear viscoelastic model that is suitable for general mechanical contact and collision problems, especially when the coefficient of restitution is high and the energy dissipation in the collision process is relatively small. In addition, the model considers not only energy loss but also the material properties, local elastic deformation, collision speed, and other factors. This model has the advantages of convenient calculation, a fast convergence speed, and so on. It is widely used in the study of the dynamics of mechanisms with clearances [2,18,26,33,34]:

$$F_n = \frac{4\delta^n}{3(\sigma_1 + \sigma_2)} \left( \frac{R_1 R_2}{R_1 + R_2} \right)^{\frac{1}{2}} \left( 1 + \frac{3(1 - c_e^2)}{4\dot{\delta}^{(-)}} \dot{\delta} \right) \tag{8}$$

where $\sigma_1 = (1 - v_1^2)/E_1$, $\sigma_2 = (1 - v_2^2)/E_2$, $v_1$ and $v_2$ are the Poisson's ratio, $E_1$ and $E_2$ are the elastic modulus, $c_e$ is the recovery coefficient, and $\dot{\delta}^{(-)}$ is the initial impact velocity. When $\delta(t_n)\delta(t_{n+1}) \le 0$, $\delta(t_n) < 0$ and $\delta(t_{n+1}) > 0$, the time of collision is between $t_n$ and $t_{n+1}$ and the impact velocity at time $t_{n+1}$ is $\dot{\delta}^{(-)}$.

A modified Coulomb friction model is used for tangential friction [2,13,27,35,36]:

$$F_t = -c_f c_d F_n \frac{v_t}{|v_t|} \tag{9}$$

where $c_f$ is the friction coefficient and $c_d$ is the dynamics correction coefficient, and

$$c_d = \begin{cases} 0 \, , |v_t| < v_0 \\ \frac{|v_t| - v_0}{v_1 - v_0}, v_0 \le |v_t| \le v_1 \\ 1 \, , |v_t| > v_1 \end{cases} \, .$$

## 3. Dynamics Optimization Model of a Mechanism with Clearances

The diagram for a 2-*DOF* nine–bar mechanism is shown in Figure 3. The 2-*DOF* nine–bar mechanism is composed of crank 1 ($L_1$), rod 2 ($L_2$), rod 3 ($L_3$), crank 4 ($L_4$), frame 5 ($L_5$), rod 6 ($L_6$), a triangle plate 7 ($L_{71}$, $L_{72}$, $L_{73}$), rod 8 ($L_8$), and a slider ($S_9$). It is known that the mechanism has eight movable members (crank 1, rod 2, rod 3, crank 4, rod 6, triangle plate 7, rod 8, and the slider) and one fixed member (frame 5). The revolute pair and translational pair are lower pairs; the number of lower pairs in this mechanism is 11. According to the calculation formula for the degree of freedom (*DOF*), the *DOF* of this mechanism is:

$$F_{DOF} = 3n - 2P_L - P_H = 3 \times 8 - 2 \times 11 - 0 = 2 \tag{10}$$

where $F_{DOF}$ is the *DOF* of the mechanism, n is the total number of moving components of the mechanism, $P_L$ is the number of lower pairs, and $P_H$ is the number of higher pairs.

It can be seen that the *DOF* of the mechanism is 2. When the mechanism has two drives, the mechanism has a unique motion; that is, the slider makes a reciprocating linear motion along the guide rail. The 2-*DOF* nine–bar mechanism has the following positive motion characteristics, such as the low and stable running speed of the slider at the bottom in the dead center, rapid return characteristics, good flexibility, the strong bearing capacity of the mechanism, etc. This 2-*DOF* nine–bar mechanism could be effective when applied to the main transmission mechanism of a hybrid drive multi–link mechanical press.

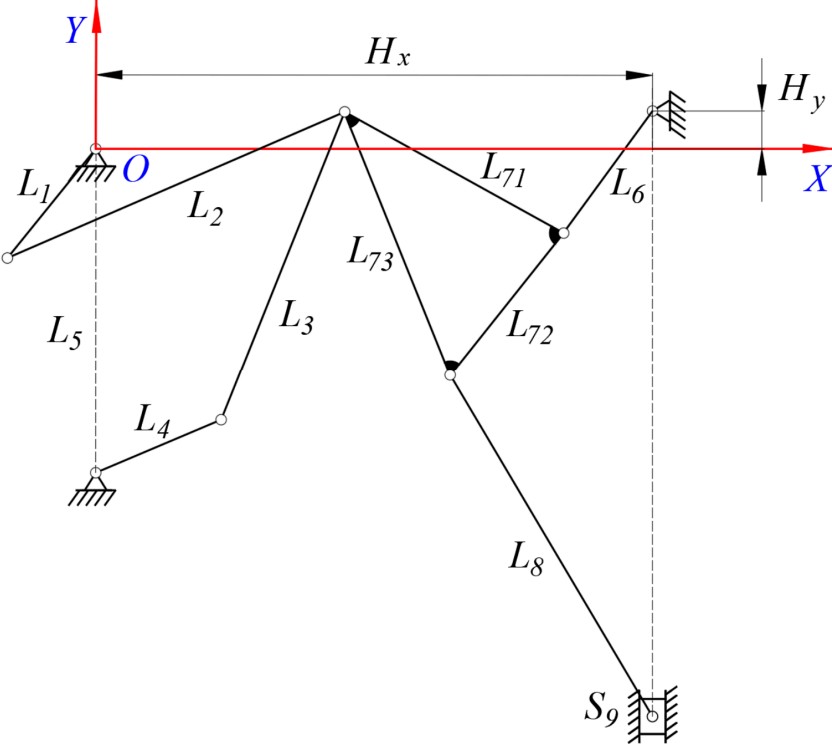

**Figure 3.** Diagram of a 2-*DOF* nine–bar mechanism.

Since the clearance of the revolute pair *A* is between crank 1 and rod 2, and the clearance of the revolute pair *B* is between crank 4 and rod 3, the impact forces at clearances *A* and *B* directly affect the dynamics of the mechanism. Consequently, this paper focuses on the influence of revolute pair clearances *A* and *B* on the mechanism. A structural diagram of the 2-*DOF* nine-bar mechanism with revolute clearances is shown in Figure 4.

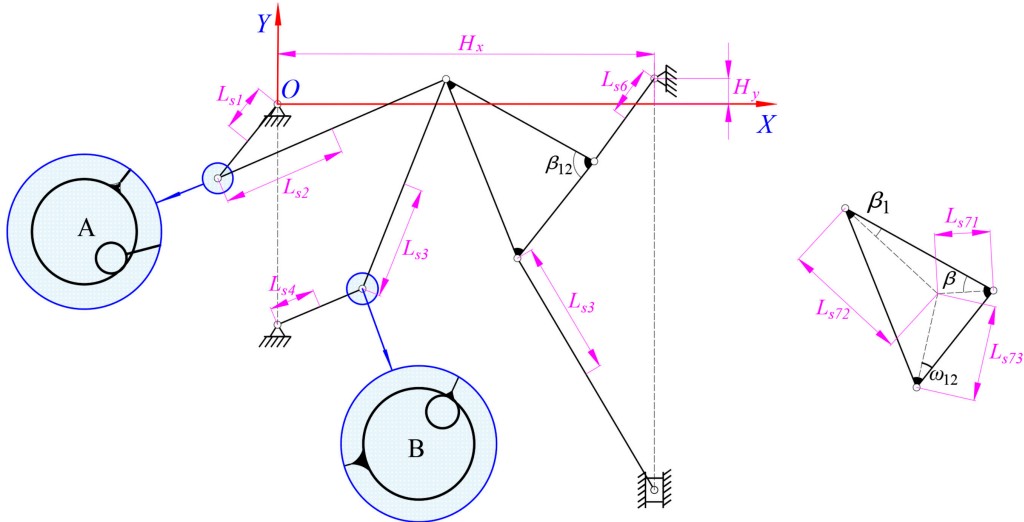

**Figure 4.** Structure diagram of a 2-*DOF* nine-bar mechanism with revolute clearances.

### 3.1. Design Variable

Since clearances *A* and *B* are located on two separate driving components, the end effector directly affects the motion characteristics of the whole mechanism, and the parameters of rod 2, rod 3, and the slider are selected for optimization. Mass, the centroid position, and the moment of inertia are important factors determining the mass distribution of components. Because rods 2 and 3 are long bars, the mass, the center of mass position,

and the moment of inertia of the rods will affect the dynamics of the mechanism. Therefore, it is necessary to optimize mass, centroid position, and the moment of inertia of rods 2 and 3. As an end effector, the slider directly affects the motion characteristics of the whole mechanism, so it needs to be optimized. Because the slider moves in translation within the guide rail, its moment of inertia has little effect on the dynamics response of the mechanism. Compared with rods 2 and 3, the length of the slider is relatively short, resulting in the relatively small influence of its center of mass position on the dynamics of the mechanism. Therefore, the quality of the slider is selected as the optimization design variable. To sum up, taking the mass, centroid position, and moment of inertia of rod 2 and rod 3 and the slider's mass as the optimization design variables:

$$
\begin{aligned}
X_c &= (m_2, L_{s2}, J_2, m_3, L_{s3}, J_3, m_9)^{\mathrm{T}} \\
&= \left(x_{(1)}, x_{(2)}, x_{(3)}, x_{(4)}, x_{(5)}, x_{(6)}, x_{(7)}\right)^{\mathrm{T}}
\end{aligned}
\tag{11}
$$

### 3.2. Constraint Conditions

The constraints of the dynamics optimization model are:

$$
X_c^L \leq X_c \leq X_c^U
\tag{12}
$$

where $X_c{}^U$ and $X_c{}^L$ are the upper and lower limits of the optimized design variables. The value ranges of $X_c{}^U$ and $X_c{}^L$ are shown in Table 1. Among them, the lower limit of component quality is half the size of the original quality, and the upper limit of component quality is twice the size of the original quality. The upper limit and lower limit of the centroid position of the member are taken as the limit positions of the member, respectively. The lower limit of the moment of inertia assumes that the rotating shaft passes through the center of mass of the rod, and the upper limit of the moment of inertia assumes that the rotating shaft passes through the end point of the rod [20,33].

**Table 1.** Value ranges of $X_c{}^U$ and $X_c{}^L$.

|  | $m_2$/kg | $L_{s2}$/mm | $J_2$/(kg·m$^2$) | $m_3$/kg | $L_{s3}$/mm | $J_3$/(kg·m$^2$) | $m_9$/kg |
|---|---|---|---|---|---|---|---|
| $X_c^L$ | $\frac{1}{2}m_2$ | 0 | $\frac{1}{12}m_2L_2{}^2$ | $\frac{1}{2}m_3$ | 0 | $\frac{1}{12}m_3L_3{}^2$ | $\frac{1}{2}m_9$ |
| $X_c^U$ | $2m_2$ | $L_2$ | $\frac{1}{3}m_2L_2{}^2$ | $2m_3$ | $L_3$ | $\frac{1}{3}m_3L_3{}^2$ | $2m_9$ |

### 3.3. Objective Functions

#### 3.3.1. Optimize the Maximum Acceleration of the Slider

As the end effector of the 2-*DOF* nine–bar mechanism, the kinematic characteristics of the slider have important research significance. Acceleration is a bridge connecting kinematics and the dynamics of the mechanism, which is closely related to the force of the system. At the same time, the peak value of the slider's acceleration intensifies the vibration due to the surge of its clearance value. Therefore, it is of great practical significance to optimize the maximum acceleration of the slider. The objective function of the optimization design is to minimize the maximum value of the slider's acceleration; the specific expression can be expressed as:

$$
F_o = \min(\|a_H\|_\infty)
\tag{13}
$$

where $a_H$ is the slider's acceleration. Here, $\|a_H\|_\infty = \max_i |a_H|$, which is the maximum of all the absolute values of slider acceleration.

#### 3.3.2. Optimize the Central Trajectory at the Clearance Joint

Due to the existence of clearance, the movement of the clearance shaft in the bearing may be random and arbitrary. Therefore, by optimizing the central trajectory at the clearance point, increasing the continuous contact state between the shaft and bearing, and

reducing the generation of the collision state, the undesirable influence of clearance on the mechanism can be reduced effectively.

Taking the minimum difference between the actual center trajectory and the ideal trajectory as the optimization objective, the specific expression of the objective function can be expressed as:

$$F_o = \min(\omega_A f_A + \omega_B f_B) \tag{14}$$

where $\omega_A$ and $\omega_B$ are the weighing factors, and expressions of $f_A$ and $f_B$ can be written as:

$$
\begin{cases}
f_A = \sqrt{(e_A^x)^2 + \left(e_A^y\right)^2 - c_A{}^2} \\
f_B = \sqrt{(e_B^x)^2 + \left(e_B^y\right)^2 - c_B{}^2}
\end{cases} \tag{15}
$$

where $e_A^x$ and $e_A^y$ are the components of eccentricity between the shaft and bearing in revolute joint $A$ in the $X$ and $Y$ directions, respectively. $e_B^x$ and $e_B^y$ are the components of eccentricity between the shaft and bearing in the revolute joint $B$ in the $X$ and $Y$ directions, respectively. $c_A$ and $c_B$ are the clearance sizes of joints $A$ and $B$, respectively.

Based on the structural diagram of the mechanism with clearances shown in Figure 1, the expressions of $e_A^x$ and $e_A^y$ can be expressed as:

$$
\begin{cases}
e_A = \left( \; x_2 - L_{s2}\cos\theta_2 - L_1\cos\theta_1 \quad y_2 - L_{s2}\sin\theta_2 - L_1\sin\theta_1 \; \right)^{\mathrm{T}} \\
\quad = \left( \; e_A^x \quad e_A^y \; \right)^{\mathrm{T}} \\
e_B = \left( \; x_3 - L_{s3}\cos\theta_3 - L_4\cos\theta_4 \quad y_3 - L_{s3}\sin\theta_3 - L_4\sin\theta_4 + L_5 \; \right)^{\mathrm{T}} \\
\quad = \left( \; e_B^x \quad e_B^y \; \right)^{\mathrm{T}}
\end{cases} \tag{16}
$$

### 3.3.3. Optimization Method

The genetic algorithm is a multi-objective optimization algorithm that can find an optimal solution by simulating the process of natural evolution. Firstly, the initial population is randomly generated as the candidate solution. Then, according to the fitness value, individuals are selected, crossed, and mutated. The cycle is repeated to obtain a group satisfying the conditions. Finally, the optimal solution of the optimal design variables is obtained.

The steps of the optimization are as follows

(1) Set the initial parameters: chromosome length, population size, maximum genetic algebra, coding type, mutation probability, crossover probability, etc;
(2) Randomly generate an initial population with size $N$, $P(0) = \left\{x_1^0, x_2^0, \cdots, x_{N-1}^0, x_N^0\right\}$;
(3) Calculate the fitness value of each individual $x_i^t (i = 1, 2, \cdots, N)$ in the population $P(t) = \left\{x_1^t, x_2^t, \cdots, x_{N-1}^t, x_N^t\right\}$;
(4) By comparing the fitness value of each individual, select individuals according to the selection operator, and keep the selected individuals with a larger fitness value until the number of the population is $N$;
(5) Optimize the combination of some genes of the surviving parent individuals, exchanging the genes at some corresponding positions of the two parent individuals according to the crossover probability, so as to produce two new individuals; the individuals that have not been crossed directly replicate into the new species group;
(6) The genes at some coding positions are mutated according to the mutation probability. The mutated individual replaces the original individual in the new species group. Individuals without compilation directly enter the new population. This completes an evolutionary process and selects the next generation of the population $P(t+1) = \left\{x_1^{t+1}, x_2^{t+1}, \cdots, x_{N-1}^{t+1}, x_N^{t+1}\right\}$;
(7) If the fitness has reached saturation or the number of iterations has reached the upper limit, it will stop. Otherwise, go to step 3.

### 3.4. Establishment of a Dynamics Optimization Model of a 2-DOF Nine-Bar Mechanism with Multiple Clearances

The generalized coordinates of each component are set as:

$$q_i = \begin{pmatrix} x_i & y_i & \theta_i \end{pmatrix}^{\mathrm{T}} \tag{17}$$

where $x_i$ and $y_i$ are the position components of component $i$ in the $X$ and $Y$ directions in the system coordinate system, and $\theta_i$ is the rotation angle of member $i$.

The 2-*DOF* nine-bar mechanism has 24 generalized coordinates; the generalized coordinates of the mechanism can be expressed as:

$$q = \begin{pmatrix} x_1 \ y_1 \ \theta_1 \ x_2 \ y_2 \ \theta_2 \ x_3 \ y_3 \ \theta_3 \ x_4 \ y_4 \ \theta_4 \cdots \\ x_6 \ y_6 \ \theta_6 \ x_7 \ y_7 \ \theta_7 \ x_8 \ y_8 \ \theta_8 \ x_9 \ y_9 \ \theta_9 \end{pmatrix}^{\mathrm{T}} \tag{18}$$

The dynamics optimization model of the mechanism with clearances is the set of the dynamics model and the optimization design variables. A displacement constraint equation with multiple clearances, including optimization design variables, could be written as:

$$\Phi(q,t) = \begin{pmatrix}
x_1 - L_{s1}\cos\theta_1 \\
y_1 - L_{s1}\sin\theta_1 \\
x_4 - L_{s4}\cos\theta_4 \\
y_4 - L_{s4}\sin\theta_4 + L_5 \\
x_7 + L_{s72}\cos(\alpha_1) - x_2 - (L_2 - x_{(2)})\cos\theta_2 \\
y_7 + L_{s72}\sin(\alpha_1) - y_2 - (L_2 - x_{(2)})\sin\theta_2 \\
x_7 + L_{s72}\cos(\alpha_1) - x_3 - (L_3 - x_{(5)})\cos\theta_3 \\
y_7 + L_{s72}\sin(\alpha_1) - y_3 - (L_3 - x_{(5)})\sin\theta_3 \\
x_6 - L_{s6}\cos\theta_6 - H_x \\
y_6 - L_{s6}\sin\theta_6 - H_y \\
x_7 - L_{s71}\cos(\alpha_2) - x_6 - L_{s6}\cos\theta_6 \\
y_7 - L_{s71}\sin(\alpha_2) - y_6 - L_{s6}\sin\theta_6 \\
x_7 + L_{s73}\cos(\alpha_3) - x_8 + L_{s8}\cos\theta_8 \\
y_7 + L_{s73}\sin(\alpha_3) - y_8 + L_{s8}\sin\theta_8 \\
x_9 - H_x - L_6\cos\theta_6 - L_{72}\cos(\alpha_4) - L_{s8}\cos\theta_8 \\
y_9 - H_y - L_6\sin\theta_6 - L_{72}\sin(\alpha_4) - L_{s8}\sin\theta_8 \\
x_9 - H_x \\
\theta_9 - 1.5\pi \\
\theta_1 - \omega_1 t - 5.7645 \\
\theta_4 - \omega_4 t + 2.4934
\end{pmatrix} = 0 \tag{19}$$

where the values of $\beta$, $\beta_1$, $\beta_{12,}$ and $\omega_{12}$ are $34.11°$, $22.22°$, $83.24°$, and $36.19°$, respectively. $H_x$ and $H_y$ are 0.08 m and 0.645 m, respectively. $\alpha_1 = \theta_7 - \beta_1$, $\alpha_2 = \theta_7 + \beta$, $\alpha_3 = \theta_7 + \beta_{12} + \omega_{12}$, $\alpha_4 = \theta_7 + \beta_{12}$.

Velocity is a constraint equation that can be expressed as [37]:

$$\Phi_q \dot{q} = -\Phi_t \equiv v \tag{20}$$

where $\Phi_q$ is the Jacobian matrix, $\Phi_q = \partial\Phi/\partial q$, $\dot{q}$ represents the generalized velocity vector, $\Phi_t = \partial\Phi/\partial t$, and $\Phi_t$ can be written as

$$\Phi_t = \frac{\partial\Phi}{\partial t} = (\mathbf{0}_{1\times18}, -\omega_1, -\omega_4)^{\mathrm{T}} \tag{21}$$

The Jacobian matrix of the mechanism can be expressed as:

$$\Phi_q = \left(\Phi_{q_{(1)}}, \Phi_{q_{(2)}}, \Phi_{q_{(3)}}, \cdots, \Phi_{q_{(22)}}, \Phi_{q_{(23)}}, \Phi_{q_{(24)}}\right) \tag{22}$$

where $\Phi_{q(1)} = (1, \mathbf{0}_{19\times1})^{\mathrm{T}}$, $\Phi_{q(2)} = (0, 1, \mathbf{0}_{18\times1})^{\mathrm{T}}$, $\Phi_{q(3)} = (L_{s1}\sin\theta_1, -L_{s1}\cos\theta_1, \mathbf{0}_{16\times1}, 1, 0)^{\mathrm{T}}$, $\Phi_{q(4)} = (\mathbf{0}_{4\times1}, -1, \mathbf{0}_{15\times1})^{\mathrm{T}}$, $\Phi_{q(5)} = (\mathbf{0}_{5\times1}, -1, \mathbf{0}_{14\times1})^{\mathrm{T}}$, $\Phi_{q(6)} = (\mathbf{0}_{4\times1}, L_{s2}\sin\theta_2, -L_{s1}\cos\theta_2, \mathbf{0}_{14\times1})^{\mathrm{T}}$, $\Phi_{q(7)} = (\mathbf{0}_{6\times1}, -1, \mathbf{0}_{13\times1})^{\mathrm{T}}$, $\Phi_{q(8)} = (\mathbf{0}_{7\times1}, -1, \mathbf{0}_{12\times1})^{\mathrm{T}}$, $\Phi_{q(9)} = (\mathbf{0}_{6\times1}, L_{s3}\sin\theta_3, -L_{s3}\cos\theta_3, \mathbf{0}_{12\times1})^{\mathrm{T}}$, $\Phi_{q(10)} = (\mathbf{0}_{2\times1}, 1, \mathbf{0}_{17\times1})^{\mathrm{T}}$, $\Phi_{q(11)} = (\mathbf{0}_{3\times1}, 1, \mathbf{0}_{16\times1})^{\mathrm{T}}$, $\Phi_{q(12)} = (\mathbf{0}_{2\times1}, L_{s4}\sin\theta_4, -L_{s4}\cos\theta_4, \mathbf{0}_{15\times1}, 1)^{\mathrm{T}}$, $\Phi_{q(13)} = (\mathbf{0}_{8\times1}, 1, 0, -1, \mathbf{0}_{9\times1})^{\mathrm{T}}$, $\Phi_{q(14)} = (\mathbf{0}_{9\times1}, 1, 0, -1, \mathbf{0}_{8\times1})^{\mathrm{T}}$, $\Phi_{q(15)} = (\mathbf{0}_{8\times1}, L_{s6}\sin\theta_6, -L_{s6}\cos\theta_6, L_{s6}\sin\theta_6, -L_{s6}\cos\theta_6, \mathbf{0}_{2\times1}, L_{s6}\sin\theta_6, -L_{s6}\cos\theta_6, \mathbf{0}_{4\times1})^{\mathrm{T}}$, $\Phi_{q(16)} = (\mathbf{0}_{4\times1}, 1, 0, 1, \mathbf{0}_{3\times1}, 1, 0, 1, \mathbf{0}_{7\times1})^{\mathrm{T}}$, $\Phi_{q(17)} = (\mathbf{0}_{5\times1}, 1, 0, 1, \mathbf{0}_{3\times1}, 1, 0, 1, \mathbf{0}_{6\times1})^{\mathrm{T}}$,

$$\Phi_{q(18)} = \begin{pmatrix} \mathbf{0}_{4\times1}, -L_{s72}\sin(\alpha_1), L_{s72}\cos(\alpha_1), -L_{s72}\sin(\alpha_1), L_{s72}\cos(\alpha_1), \mathbf{0}_{2\times1}, \\ L_{s71}\sin(\alpha_2), \cdots \\ -L_{s71}\cos(\alpha_2), -L_{s73}\sin(\alpha_3), L_{s73}\cos(\alpha_3), L_{72}\sin(\alpha_4), -L_{72}\cos(\alpha_4), \mathbf{0}_{4\times1} \end{pmatrix}^{\mathrm{T}},$$

$\Phi_{q(19)} = (\mathbf{0}_{12\times1}, -1, \mathbf{0}_{7\times1})^{\mathrm{T}}$, $\Phi_{q(20)} = (\mathbf{0}_{13\times1}, -1, \mathbf{0}_{6\times1})^{\mathrm{T}}$, $\Phi_{q(21)} = (\mathbf{0}_{12\times1}, -L_{s8}\sin\theta_8, L_{s8}\cos\theta_8, L_{s8}\sin\theta_8, -L_{s8}\cos\theta_8, \mathbf{0}_{4\times1})^{\mathrm{T}}$, $\Phi_{q(22)} = (\mathbf{0}_{14\times1}, 1, 0, 1, \mathbf{0}_{3\times1})^{\mathrm{T}}$, $\Phi_{q(23)} = (\mathbf{0}_{15\times1}, 1, \mathbf{0}_{4\times1})^{\mathrm{T}}$, $\Phi_{q(24)} = (\mathbf{0}_{17\times1}, 1, \mathbf{0}_{2\times1})^{\mathrm{T}}$.

The acceleration constraint equations could be expressed as [37]:

$$\Phi_q\ddot{q} = -\left(\Phi_q\dot{q}\right)_q\dot{q} - 2\Phi_{qt}\dot{q} - \Phi_{tt} \equiv \gamma \tag{23}$$

where $\ddot{q}$ represents the generalized acceleration vector, $\Phi_{qt}$ is the partial derivative of the Jacobian matrix with respect to time, $\Phi_{qt} = 0_{20\times24}$, $\Phi_{tt}$ is the partial derivative of $\Phi_t$ with respect to time, $\Phi_{tt} = 0_{20\times1}$, and $\gamma$ is the vector of quadratic velocity terms.

$\Phi_{qq}$ can be written as:

$$\Phi_{qq} = \left(\Phi_{q(1)q(1)}, \Phi_{q(2)q(2)}, \Phi_{q(3)q(3)}, \cdots, \Phi_{q(22)q(22)}, \Phi_{q(23)q(23)}, \Phi_{q(24)q(24)}\right) \tag{24}$$

where: $\Phi_{q(1)q(1)} = \Phi_{q(2)q(2)} = \Phi_{q(4)q(4)} = \Phi_{q(5)q(5)} = \Phi_{q(7)q(7)} = \Phi_{q(8)q(8)} = \Phi_{q(10)q(10)} = \Phi_{q(11)q(11)} = \Phi_{q(13)q(13)} = \Phi_{q(14)q(14)} = \Phi_{q(16)q(16)} = \Phi_{q(17)q(17)} = \Phi_{q(19)q(19)} = \Phi_{q(20)q(20)} = \Phi_{q(22)q(22)} = \Phi_{q(23)q(23)} = \Phi_{q(24)q(24)} = 0_{1\times20}$, $\Phi_{q(3)q(3)} = (L_{s1}\omega_1\cos\theta_1, L_{s1}\omega_1\sin\theta_1, \mathbf{0}_{18\times1})^{\mathrm{T}}$, $\Phi_{q(6)q(6)} = (\mathbf{0}_{4\times1}, L_{s2}\omega_2\cos\theta_2, L_{s1}\omega_2\sin\theta_2, \mathbf{0}_{14\times1})^{\mathrm{T}}$, $\Phi_{q(9)q(9)} = (\mathbf{0}_{6\times1}, L_{s3}\omega_3\cos\theta_3, L_{s3}\omega_3\sin\theta_3, \mathbf{0}_{12\times1})^{\mathrm{T}}$, $\Phi_{q(12)q(12)} = (\mathbf{0}_{2\times1}, L_{s4}\omega_4\cos\theta_4, L_{s4}\omega_4\sin\theta_4, \mathbf{0}_{16\times1})^{\mathrm{T}}$, $\Phi_{q(15)q(15)} = (\mathbf{0}_{8\times1}, L_{s6}\omega_6\cos\theta_6, L_{s6}\omega_6\sin\theta_6, L_{s6}\omega_6\cos\theta_6, L_{s6}\omega_6\sin\theta_6, \mathbf{0}_{3\times1}, L_{s6}\omega_6\cos\theta_6, L_{s6}\omega_6\sin\theta_6, \mathbf{0}_{4\times1})^{\mathrm{T}}$,

$$\Phi_{q(18)q(18)} = \begin{pmatrix} \mathbf{0}_{4\times1}, -L_{s72}\omega_7\cos(\alpha_1), -L_{s72}\omega_7\sin(\alpha_1), -L_{s72}\omega_7\cos(\alpha_1), -L_{s72}\omega_7 \\ \sin(\alpha_1), \mathbf{0}_{2\times1}, L_{s71}\omega_7\cos(\alpha_2), \cdots \\ L_{s71}\omega_7\sin(\alpha_2), -L_{s73}\omega_7\cos(\alpha_3), -L_{s73}\omega_7\sin(\alpha_3), L_{72}\omega_7\cos(\alpha_4), \\ L_{72}\omega_7\sin(\alpha_4), \mathbf{0}_{4\times1} \end{pmatrix}^{\mathrm{T}},$$

$\Phi_{q(21)q(21)} = (\mathbf{0}_{12\times1}, -L_{s8}\omega_8\cos\theta_8, -L_{s8}\omega_8\sin\theta_8, L_{s8}\omega_8\cos\theta_8, L_{s8}\omega_8\sin\theta_8, \mathbf{0}_{4\times1})^{\mathrm{T}}$.

The system dynamics equation considering optimal design variables can be expressed as follows [38]:

$$M\ddot{q} + \Phi_q^{\mathrm{T}}\lambda = g \tag{25}$$

where $M$ is the mass matrix of the system, $\lambda$ is the Lagrange multiplier, and $g$ is the generalized force of the system, including gravity, external force, and external torque.

A quality matrix considering the optimal design variables of a 2-*DOF* nine–bar mechanism with clearances can be expressed as:

$$M = \mathrm{diag}\begin{pmatrix} m_1 \; m_1 \; J_1 \; x_{(1)} \; x_{(1)} \; x_{(3)} \; x_{(4)} \; x_{(4)} \; x_{(6)} \; m_4 \; m_4 \; J_4 \cdots \\ m_6 \; m_6 \; J_6 \; m_7 \; m_7 \; J_7 \; m_8 \; m_8 \; J_8 \; x_{(7)} \; x_{(7)} \; J_9 \end{pmatrix}. \tag{26}$$

The generalized force of a system with optimal design variables can be expressed as:

$$
g = \begin{pmatrix} F_x^A, F_y^A, M_{A1}, -F_x^A, -F_y^A, M_{A2}, F_x^B, F_y^B, M_{B1}, \cdots \\ -F_x^B, -F_y^B, M_{B2}, 0, 0, 0, 0, 0, 0, 0, 0, 0, 0, 0, 0, 0 \end{pmatrix} \tag{27}
$$

$$
\begin{cases}
M_{A1} = -\left(OQ_1^y - y_1\right)F_A^x + (OQ_1^x - x_1)F_A^y \\
M_{A2} = -\left(OQ_2^y - y_2\right)F_A^x + (OQ_2^x - x_2)F_A^y \\
M_{B1} = -\left(OQ_4^y - y_4\right)F_B^x + (OQ_4^x - x_4)F_B^y \\
M_{B2} = -\left(OQ_3^y - y_3\right)F_B^x + (OQ_3^x - x_3)F_B^y
\end{cases} \tag{28}
$$

where $F_i^j (i = A, B; j = X, Y)$ are the components of contact force in the $j$ directions at clearance joint $i$, respectively. $Q_1$ and $Q_2$ are the collision points of the bearing and shaft in rotating pair $A$, respectively. $Q_3$ and $Q_4$ are the collision points of the bearing and shaft in rotating pair $B$, respectively:

$$
\begin{cases}
OQ_1 = OP_1 + R_1 n_A \\
OQ_2 = OP_2 + R_2 n_A \\
OQ_3 = OP_3 + R_2 n_B \\
OQ_4 = OP_4 + R_1 n_B
\end{cases} \tag{29}
$$

where $n_A$ and $n_B$ are unit normal vectors of clearances $A$ and $B$, respectively. $P_1$ and $P_2$ are the center points of the bearing and shaft in revolute $A$, respectively. $P_3$ and $P_4$ are the center points of the bearing and shaft in revolute $A$, respectively:

$$
\begin{cases}
OP_1 = \begin{pmatrix} L_1 \cos\theta_1 & L_1 \sin\theta_1 \end{pmatrix} \\
OP_2 = \begin{pmatrix} x_2 - x_{(2)}\cos\theta_2 & y_2 - x_{(2)}\sin\theta_2 \end{pmatrix} \\
OP_3 = \begin{pmatrix} x_3 - x_{(5)}\cos\theta_3 & y_3 - x_{(5)}\sin\theta_3 \end{pmatrix} \\
OP_4 = \begin{pmatrix} L_4 \cos\theta_4 & L_4 \sin\theta_4 & -L_5 \end{pmatrix}
\end{cases} . \tag{30}
$$

The dynamics equation of the mechanism can be obtained by solving Equations (23) and (25), and the equation expression is as follows:

$$
\begin{pmatrix} M & \Phi_q^T \\ \Phi_q & 0 \end{pmatrix} \begin{pmatrix} \ddot{q} \\ \lambda \end{pmatrix} = \begin{pmatrix} g \\ \gamma \end{pmatrix}. \tag{31}
$$

In order to overcome the default of the dynamics equation, based on the Baumgarte default stability algorithm, a strong nonlinear equation with optimal design variables is solved [39–41]:

$$
\begin{pmatrix} M & \Phi_q^T \\ \Phi_q & 0 \end{pmatrix} \begin{pmatrix} \ddot{q} \\ \lambda \end{pmatrix} = \begin{pmatrix} g \\ \gamma - 2\alpha\dot{\Phi} - \beta^2\Phi \end{pmatrix} \tag{32}
$$

where $\alpha$ and $\beta$ are the correction parameters, and $\dot{\Phi} = \frac{d\Phi}{dt}$.

Because the dynamics equation of the mechanism containing the clearances is a very nonlinear equation, it is difficult to solve the equation stably. Variable-order numerical differentiation algorithms (NDFs) are used to solve the dynamics equations, while the high-order equations are transformed into low-order equations. Therefore, the ode15s function in MATLAB is used to solve Equation (32) and convert the second-order differential equation into a first-order differential equation:

$$
\begin{cases}
\frac{d\dot{q}_1}{dt} = g_1, \frac{dq_1}{dt} = \dot{q}_1 \\
\frac{d\dot{q}_2}{dt} = g_2, \frac{dq_2}{dt} = \dot{q}_2 \\
\qquad\vdots \\
\frac{d\dot{q}_i}{dt} = g_i, \frac{dq_i}{dt} = \dot{q}_i
\end{cases} \tag{33}
$$

where $q_i$ represents the generalized coordinates and $i$ is the number of generalized coordinates.

## 4. Establishment of a Dynamics Accuracy and Reliability Model

It is assumed that the error function of the mechanism is $h(x,t) = P - P'$, where $P$ is the actual situation of the dynamics c response of the mechanism, and $P'$ is the ideal situation of the dynamics response of the mechanism. According to the error characteristics of random variables, while $|h(x,t)| \leq |\varepsilon|$, the mechanism is reliable; if not, the mechanism fails.

For mechanism reliability, the actual error is stress, and the allowable error is strength. Assuming that these are the actual and allowable errors of the motion characteristics, where driving torque meets a normal distribution, the reliability of the mechanism can be solved according to the stress-strength interference theory. Based upon the stress-strength interference theory, reliability is defined as the probability that strength is greater than stress [27]:

$$R = P\left(S - \widetilde{S} > 0\right) = P(Z > 0) \tag{34}$$

where $Z$ is the safety margin.

It is assumed that stress and strength obey a normal distribution, and the safety margin also obeys normal distribution:

$$f(Z) = \frac{1}{\sqrt{2\pi}\sigma_Z} e^{-\frac{1}{2}\left(\frac{Z-\mu_Z}{\sigma_Z}\right)^2} \tag{35}$$

where $\mu_Z = \mu_S - \mu_{\widetilde{S}}$, $\sigma_Z = \sqrt{\left(\sigma_S^2 + \sigma_{\widetilde{S}}^2\right)}$, $\mu_S$ and $\sigma_S$ are the mean value and variance of intensity, and $\mu_{\widetilde{S}}$ and $\sigma_{\widetilde{S}}$ are the mean value and the variance of stress.

Reliability could be written as:

$$P(Z > 0) = \int_0^\infty \frac{1}{\sqrt{2\pi}\sigma_Z} e^{-\frac{1}{2}\left(\frac{Z-\mu_Z}{\sigma_Z}\right)^2} dZ = \int_{z_0}^\infty \varphi(z) dz = 1 - \Phi(z_0) \tag{36}$$

where $\varphi(z) = \frac{1}{\sqrt{2\pi}} e^{-\frac{1}{2}z^2}$, $z = \frac{Z-\mu_Z}{\sigma_Z}$, and $z_0 = -\frac{\mu_Z}{\sigma_Z} = -\frac{\mu_S - \mu_{\widetilde{S}}}{\left(\sigma_S^2 + \sigma_{\widetilde{S}}^2\right)^{\frac{1}{2}}}$.

The reliability index can be expressed as:

$$\beta = \frac{\mu_Z}{\sigma_Z} = \frac{\mu_S - \mu_{\widetilde{S}}}{\sqrt{\sigma_S^2 + \sigma_{\widetilde{S}}^2}}. \tag{37}$$

## 5. Effect of Optimization on the Dynamics Responses and the Dynamics Accuracy and Reliability of a Mechanism with Clearances

### 5.1. Simulation Parameters

The mass, moment of inertia, geometric parameters, and clearance parameters are shown in Tables 2 and 3.

**Table 2.** Geometric and inertial parameters.

| Component | Length of Rod (mm) | Centroid Position Length (mm) | Mass (kg) | Moment of Inertia ($10^{-3}$ kg·m²) |
|---|---|---|---|---|
| Crank 1 | 45 | 23 | 0.148 | 0.2382 |
| Rod 2 | 326 | 163 | 0.805 | 8.007 |
| Rod 3 | 497 | 248.8 | 0.603 | 13.37 |
| Crank 4 | 95 | 48 | 0.265 | 1.210 |
| Frame 5 | 430 | — | — | — |
| Rod 6 | 230 | 115 | 0.581 | 12.12 |
| Triangular panel 7 | $L_{71}$ 325 | $L_{s71}$ 147 | 38.02 | 4.334 |
| | $L_{72}$ 250 | $L_{s72}$ 219 | | |
| | $L_{73}$ 386 | $L_{s73}$ 190 | | |
| Rod 8 | 335 | 167.6 | 0.827 | 8.663 |
| Slider 9 | — | — | 0.801 | — |

**Table 3.** Clearance simulation parameters.

| Parameter | Parameter Values | Parameter | Parameter Values |
|---|---|---|---|
| Bearing radius $R_1$ | 15 (mm) | Clearance value | 0.1 (mm) |
| Restitution coefficient $c_e$ | 0.9 | Speed of crank 1 | $2\pi$ rad/s |
| Poisson ratio $\nu_1$, $\nu_2$ | 0.3 | Speed of crank 4 | $-2\pi$ rad/s |
| Elastic modulus $E_1$, $E_2$ | 207 (GPa) | Integral tolerance | 0.000001 |
| Friction coefficient | 0.01 | Integral step | 0.0001 s |

In order to exaggerate the influence of the clearance parameters on the dynamics response of the mechanism, the clearance value at revolute pairs A and B is set as 0.1 mm, and the friction coefficient at the clearance is set as 0.01, which will lead to a sharp increase in the peak value of the dynamics response.

When the optimization objective is to minimize the maximum acceleration of the slider, rod 2, rod 3, and the slider are optimized according to the principle of mass distribution. The optimization results are as follows:

$$
\begin{cases}
m_2 = 1.574 \ (\text{kg}) \\
L_{s2} = 0.241 \ (\text{m}) \\
J_2 = 0.007 \ (\text{kg} \cdot \text{m}^2) \\
m_3 = 1.12 \ (\text{kg}) \\
L_{s3} = 0.235 \ (\text{m}) \\
J_3 = 0.033 \ (\text{kg} \cdot \text{m}^2) \\
m_9 = 1.574 \ (\text{kg})
\end{cases}
. \tag{38}
$$

When the optimization objective is to minimize the difference between the actual center trajectory and the ideal trajectory, the optimization results are as follows:

$$
\begin{cases}
m_2 = 0.538 \ (\text{kg}) \\
L_{s2} = 0.244 \ (\text{m}) \\
J_2 = 0.004 \ (\text{kg} \cdot \text{m}^2) \\
m_3 = 0.862 \ (\text{kg}) \\
L_{s3} = 0.319 \ (\text{m}) \\
J_3 = 0.006 \ (\text{kg} \cdot \text{m}^2) \\
m_9 = 0.4 \ (\text{kg})
\end{cases}
. \tag{39}
$$

*5.2. Comparative Analysis of Dynamics Behavior before and after Optimization*

5.2.1. Comparative Analysis of Dynamics Response of the Mechanism before and after Optimization

Based on Equations (26) and (27), the dynamics response of the mechanism after stabilization is shown in Figure 5. When the optimization's objective function is to minimize the maximum acceleration of the slider, it is characterized by a blue line in Figure 5 and is named "Optimization 1". When the optimization objective function is to minimize the difference between the actual trajectory and ideal trajectory, it is characterized by a green line in Figure 4 and is named "Optimization 2".

According to the acceleration curve, there are two optimization methods that can both effectively reduce the peak value of acceleration. It shows that the two methods have a good effect on improving the slider's acceleration, but the optimization effect in "Optimization 2" is better than that of "Optimization 1". Similarly, it can be seen from the contact force at the clearance joint and center trajectory diagram that when "Optimization 2" is adopted, the optimization effect is significant, and the times of peak and large amplitude of the dynamics are significantly reduced. However, when "Optimization 1" is adopted, the peak value of the contact force at the clearance joint is sometimes smaller than it was before optimization, and sometimes increases sharply. To sum up, when the objective function

of optimization is to minimize the difference between the actual trajectory and the ideal trajectory, the optimization effect is stronger than that when the optimization objective function is to minimize the maximum acceleration of the slider.

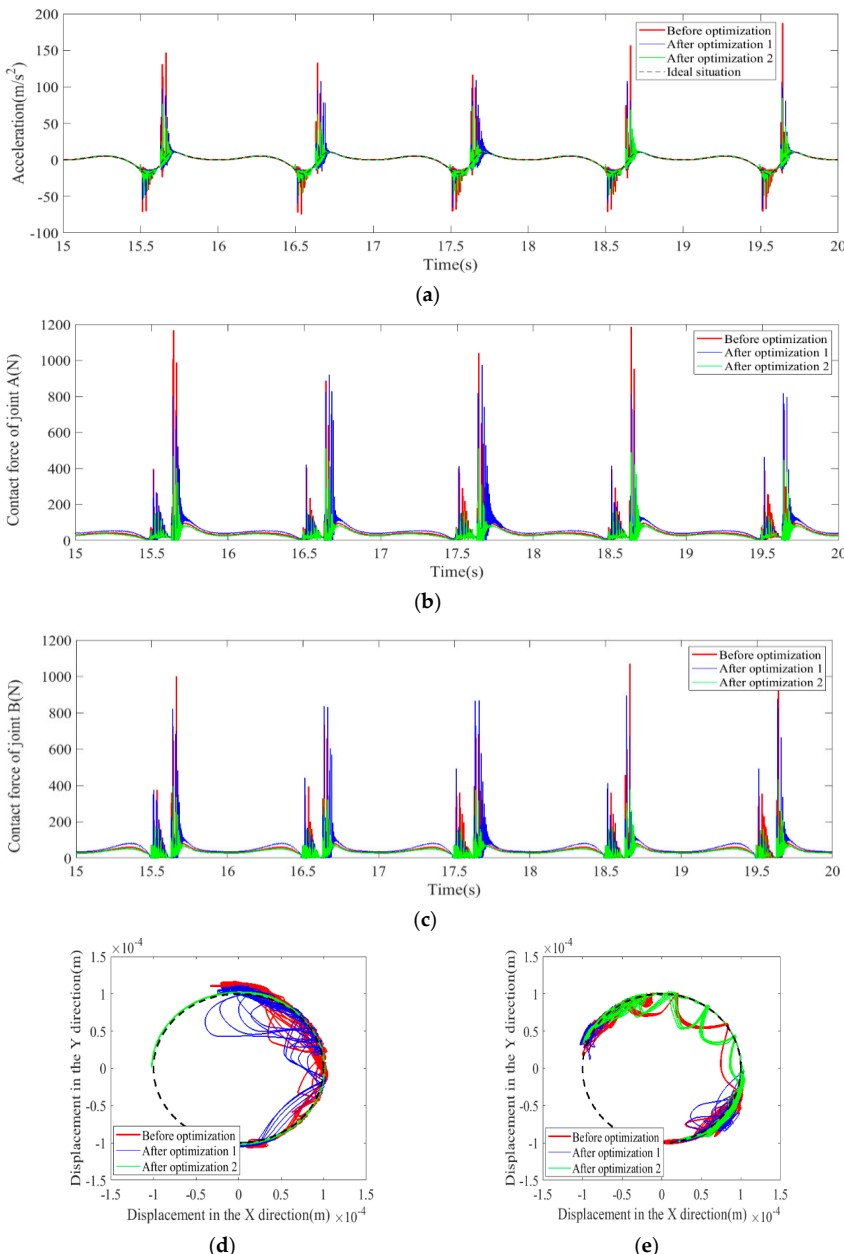

**Figure 5.** Comparative analysis of dynamics response before and after optimization. (**a**) Acceleration. (**b**) Contact force of joint A. (**c**) Contact force of joint B. (**d**) central trajectory of clearance A. (**e**) central trajectory of clearance B.

One of the cycles of data is selected for analysis, and the effects of the different optimization objective functions on the dynamics responses of the mechanism are analyzed numerically. The diagram for the dynamics response is shown in Figure 6. It can be seen from the diagram that clearances have little effect on displacement and velocity, which is almost close to an ideal state. When the mechanism has not been optimized, the peak value of the acceleration of the slider, the contact force of joint A, the contact force of joint B, the driving torque of crank 1, and the driving torque of crank 4 are 146.9 m/s², 1157 N, 1001 N, −70.09 N·m, and 52 N·m, respectively. When the objective function of optimization is to minimize the maximum acceleration of the slider, the corresponding peak values are

98.13 m/s$^2$, 806 N, 820.2 N, −52.29 N·m, and 36.11 N·m, respectively. When the objective function of optimization is to minimize the difference between the actual trajectory and ideal trajectory, the corresponding peak values are 76.81 m/s$^2$, 469.4 N, 395.7 N, −26.11 N·m, and 20.95 N·m, respectively. When the objective function of optimization is to minimize the maximum acceleration of the slider, the optimization efficiencies are 33.20%, 30.34%, 18.06%, 25.40%, and 30.56%, respectively. When the objective function of optimization is to minimize the difference between the actual trajectory and ideal trajectory, the corresponding optimization efficiencies are 47.71%, 59.43%, 60.47%, 62.75%, and 59.71%, respectively. It can be seen that the optimization effect of "Optimization 2" is significantly better than that of "Optimization 1".

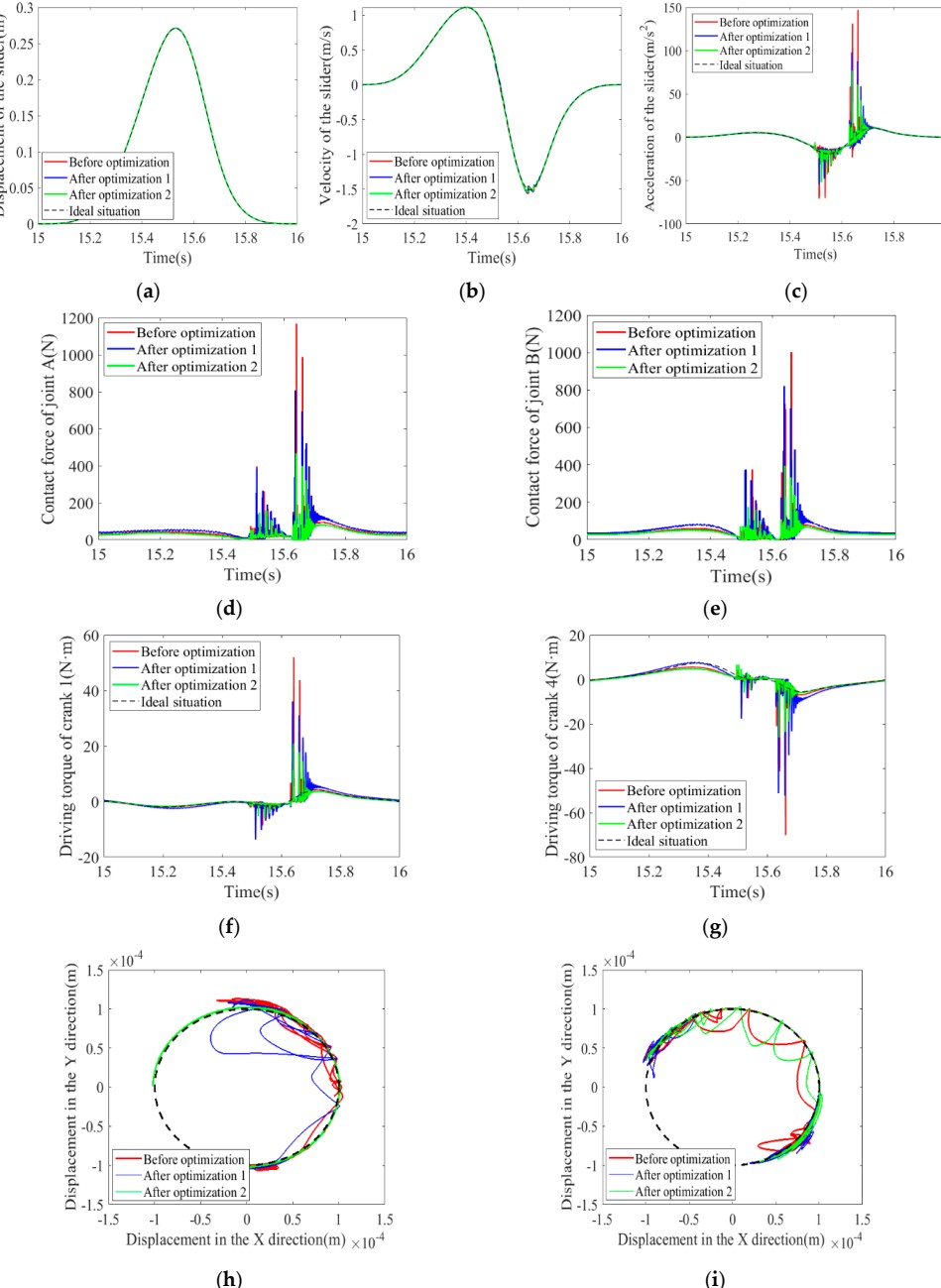

**Figure 6.** Dynamics response. (**a**) Displacement of the slider. (**b**) Velocity of the slider. (**c**) Acceleration of the slider. (**d**) Contact force of joint A. (**e**) Contact force of joint B. (**f**) Driving torque of crank 1. (**g**) Driving torque of crank 4. (**h**) Central trajectory of clearance A. (**i**) Central trajectory of clearance B.

### 5.2.2. Comparative Analysis of the Nonlinear Characteristics of the Mechanism before and after Optimization

Based on the optimization data obtained by optimizing the central trajectory at the clearance joints, the nonlinear characteristics of the clearance before and after optimization are analyzed. When the mechanism runs for 500 cycles, the nonlinear characteristics of a mechanism containing clearances before and after optimization are analyzed with a phase map and Poincaré map. The nonlinear characteristics of joints A and B are shown in Figures 7–10. The nonlinear characteristics of the slider's error are shown in Figure 11.

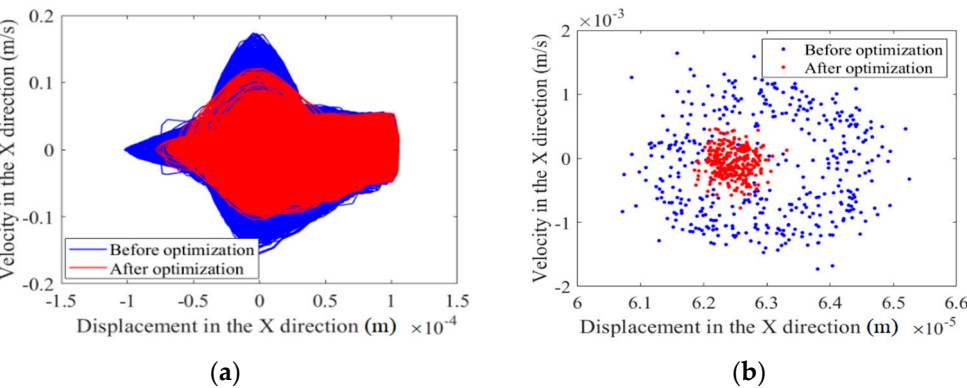

(**a**)  (**b**)

**Figure 7.** Nonlinear characteristics of joint A in the *X* direction. (**a**) Phase map. (**b**) Poincaré map.

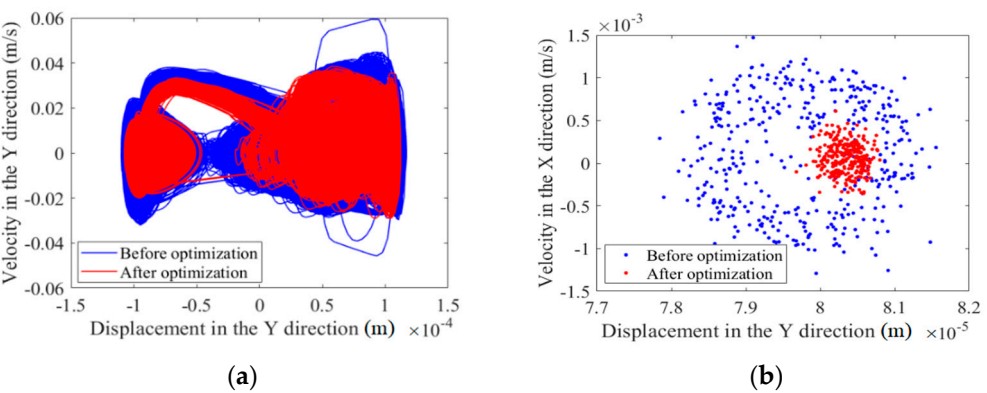

(**a**)  (**b**)

**Figure 8.** Nonlinear characteristics of joint A in the *Y* direction. (**a**) Phase map. (**b**) Poincaré map.

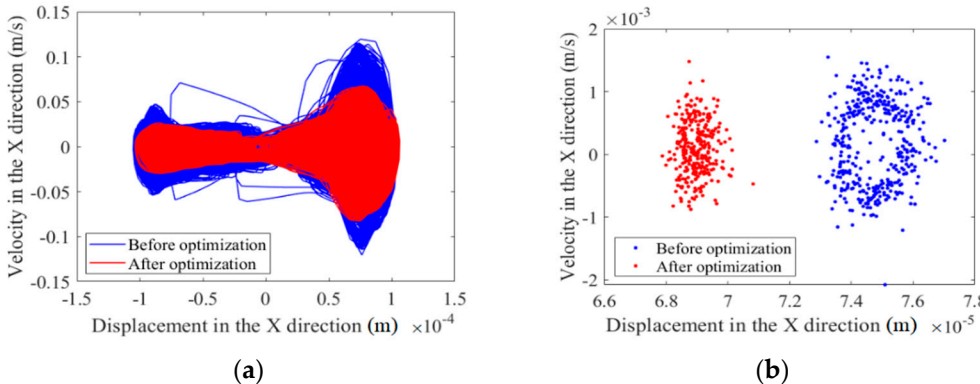

(**a**)  (**b**)

**Figure 9.** Nonlinear characteristics of joint B in the *X* direction. (**a**) Phase map. (**b**) Poincaré map.

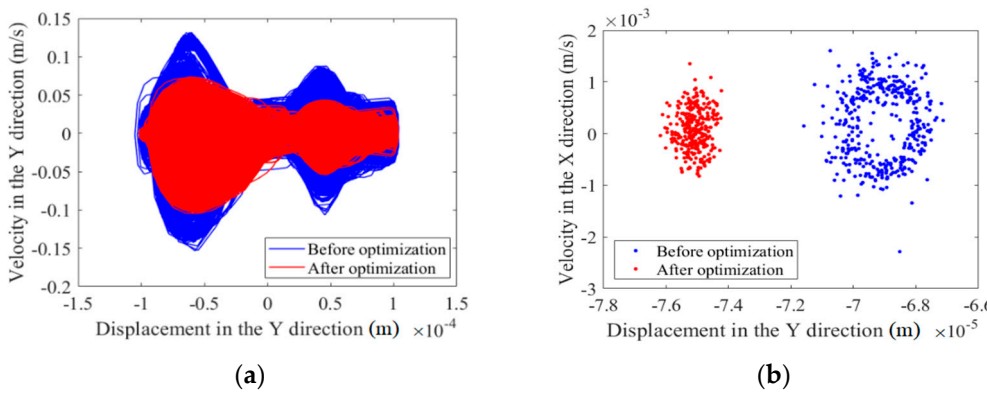

**Figure 10.** Nonlinear characteristics of joint B in the *Y* direction. (**a**) Phase map. (**b**) Poincaré map.

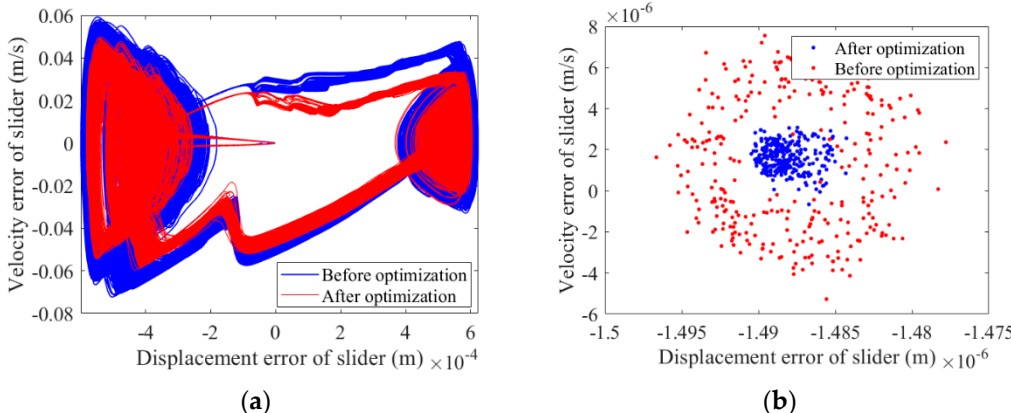

**Figure 11.** Nonlinear characteristic of the slider's error. (**a**) Phase map. (**b**) Poincaré map.

It can be seen from the phase diagrams and Poincaré maps that the area of the phase diagram after optimization decreases; the dispersion of mapping points after optimization weakens and becomes relatively concentrated. It can be seen that clearance optimization can effectively improve both the chaos and the stability of the mechanism.

*5.3. Dynamics Accuracy and Reliability Analysis of a Mechanism with Clearances before and after Optimization*

Based on the data yielded after optimization (where the optimization objective is to minimize the difference between the actual trajectory and ideal trajectory), the dynamics accuracy and reliability of the mechanism before and after optimization are compared and analyzed. The reliability index diagrams of displacement, velocity, acceleration, torque 1, and torque 4 are shown in Figure 12.

Because the clearance of the kinematic pair has little effect on the slider's displacement, the change in the reliability index of the slider's displacement is not obvious, but the reliability of the slider's displacement has been improved after optimization. The sensitivity of the slider's velocity to the clearance is stronger than that of the slider's displacement, so the reliability index of velocity is significantly improved. The sensitivity of the slider's acceleration and the driving torque to clearance are strong, dynamical response curves with an obvious vibration and peak, and the response curves have been significantly improved after optimization. Therefore, the reliability indexes of acceleration and torque are obviously larger. In general, optimization enhances the reliability of the mechanism, and the mechanism becomes more stable.

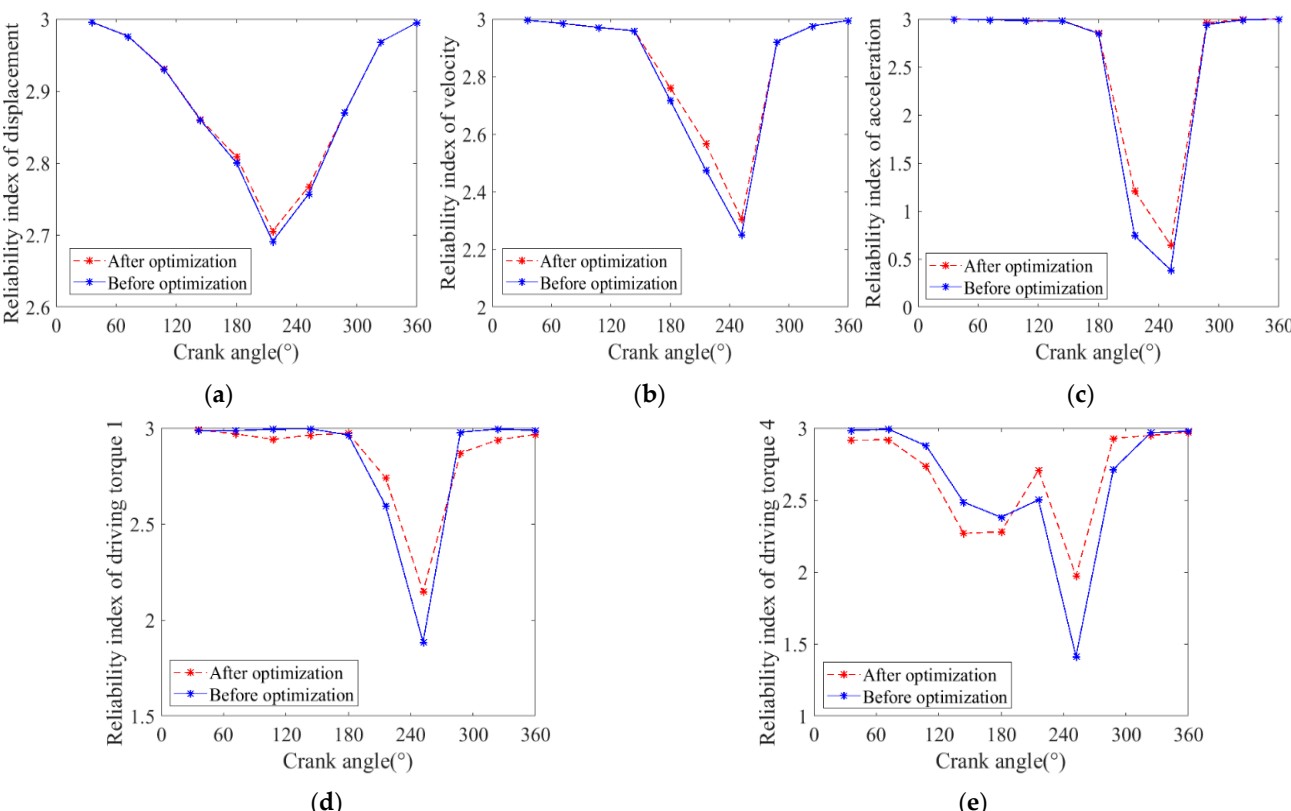

**Figure 12.** Reliability index. (**a**) Displacement. (**b**) Velocity. (**c**) Acceleration. (**d**) Driving torque of crank 1. (**e**) Driving torque of crank 4.

*5.4. Experimental Research into a Mechanism with Clearances*

Design of the Test Platform for a Mechanism with Clearances

The 2-*DOF* nine–bar mechanism test bed is shown in Figure 13. The structural diagram of the 2-*DOF* nine–bar mechanism test platform is shown in Figure 14. The two motors are installed under the support plate, and the two cranks are driven by motors to drive the slider to move back and forth along the guide rail. The clearances between the two cranks and connecting rods are considered (revolute clearances A and B). The component is made of aluminum alloy, the elastic model is 70 Gpa, the bearing radius is 12 mm, and pin shafts with a clearance value of 0.1 mm are shown in Figure 15. In order to facilitate the processing and installation of parts, we have modified the dimensions to some extent. However, the theoretical results shown in Figure 16b are based on the theoretical model proposed in this paper. The geometric and inertial parameters of the test platform are shown in Table 4.

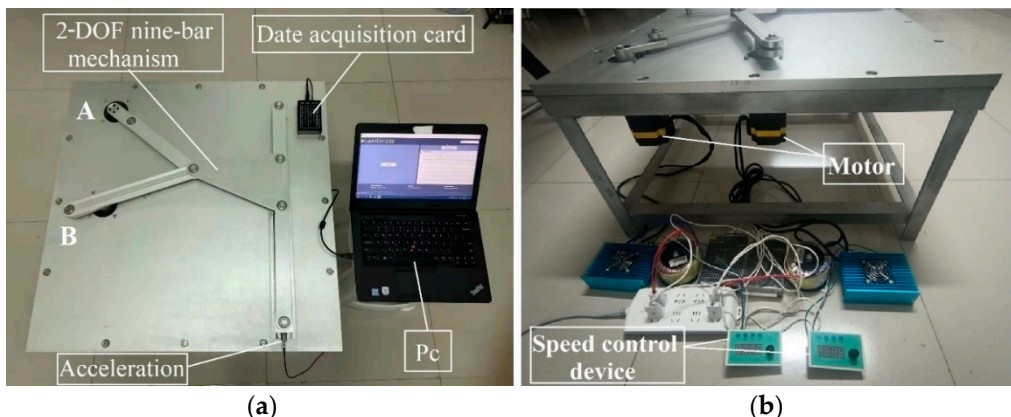

**Figure 13.** The 2-*DOF* nine–bar mechanism test bed. (**a**) Vertical view. (**b**) Lateral view.

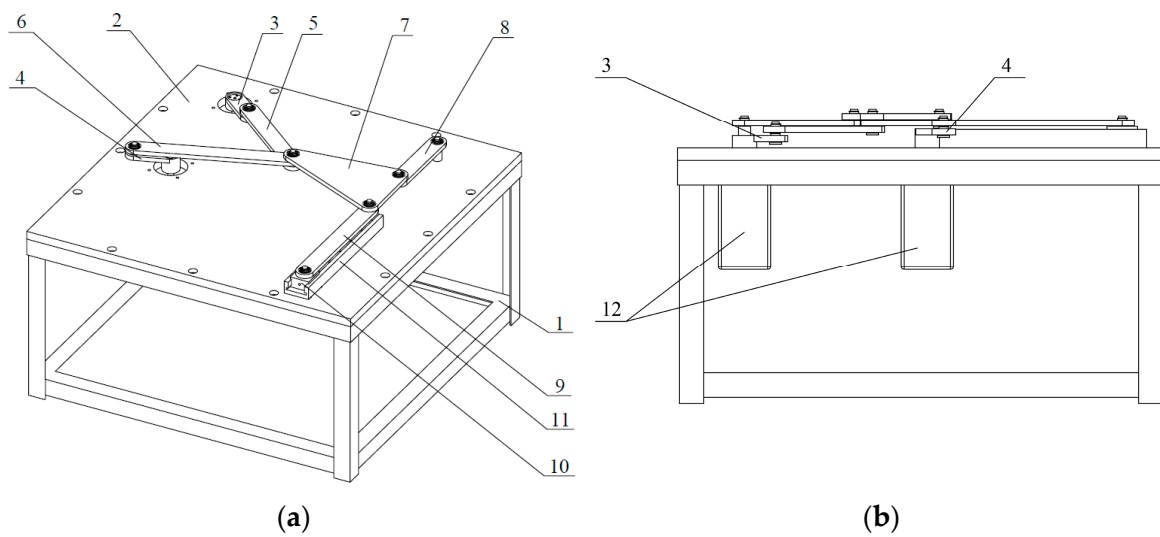

**Figure 14.** Structure diagram of the 2-*DOF* nine–bar mechanism testbed. (**a**) Normal axonometric drawing. (**b**) Lateral view. 1—Frame, 2—Support plate, 3—Crank 1, 4—Crank 4, 5—Rod 2, 6—Rod 3, 7—Triangular panel 7, 8—Rod 6, 9—Rod 8, 10—Slider, 11—Guide rail, 12—Motor.

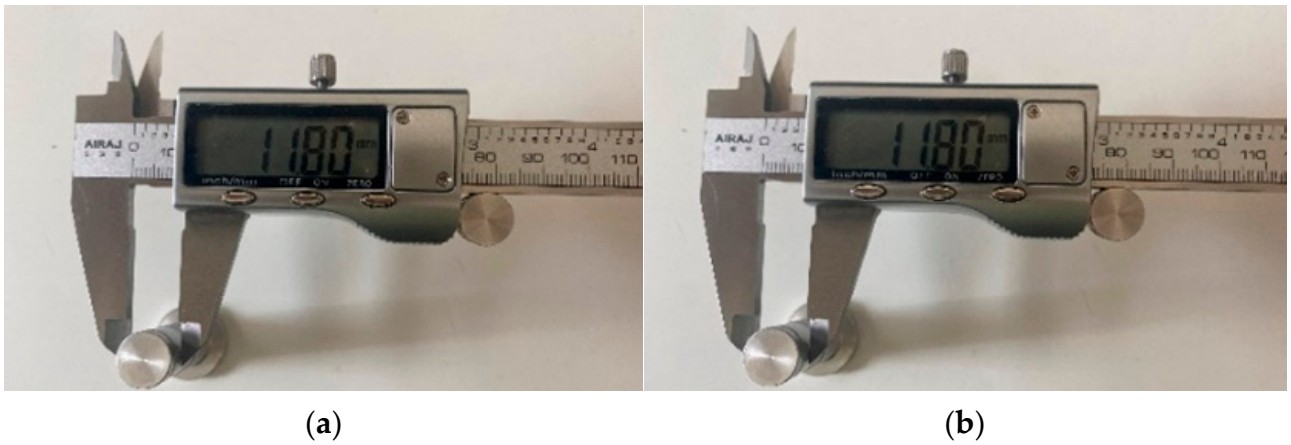

**Figure 15.** Shaft with clearance (clearance value = 0.1 mm). (**a**) Shaft A. (**b**) Shaft B.

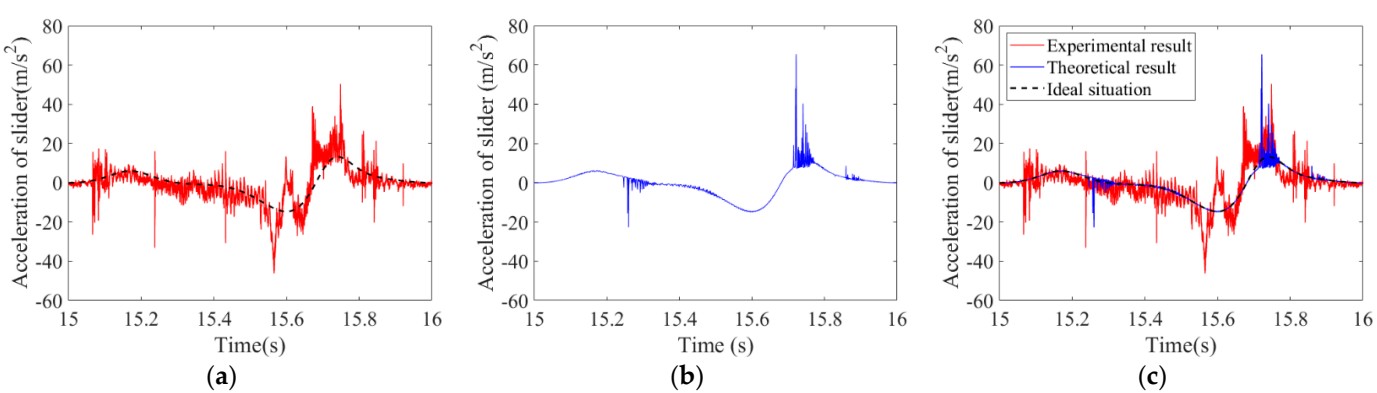

**Figure 16.** Comparison between experimental results and theoretical results. (**a**) Experimental result. (**b**) Theoretical result. (**c**) Comparison diagram.

**Table 4.** Geometric and inertial parameters of the test platform.

| Component | Length of Rod (m) | Mass (kg) | Moment of Inertia ($10^{-3}$ kg·m$^2$) |
|---|---|---|---|
| Crank 1 | 0.08 | 0.142 | 0.303 |
| Crank 4 | 0.08 | 0.142 | 0.303 |
| Rod 2 | 0.250 | 0.284 | 1.479 |
| Rod 3 | 0.350 | 0.396 | 4.043 |
| | 0.247 | | |
| Triangular panel 7 | 0.140 | 0.837 | 3.076 |
| | 0.270 | | |
| Rod 6 | 0.180 | 0.206 | 3.6863 |
| Rod 8 | 0.300 | 0.341 | 2.558 |
| Slider 9 | — | 0.112 | — |

When the operating speeds of Crank 1 and Crank 4 are 60 rpm and −60 rpm, respectively, the comparison between the experimental results and theoretical results is shown in Figure 16. According to Figure 16a, the peak value of acceleration corresponding to the experimental results is 50.36 m/s$^2$. According to Figure 16b, the peak value of acceleration corresponding to the theoretical calculation results is 65.43 m/s$^2$. As shown in Figure 16c, it can be seen from the figure that the two curves are basically consistent in trend, but there is a certain difference in value. The comparison between the theoretical and experimental numerical results of the maximum acceleration is shown in Table 5. According to the data in Table 5, the error between the theoretical result and the experimental result is 15.07 m/s$^2$, and the error rate is 23.03%. The main reason is that the vibration of the mechanism and the friction of the components lead to a decrease in the peak value of the experimental results and the acceleration of the vibration frequency. This can basically verify the correctness of the theoretical results.

**Table 5.** The comparison between the theoretical and experimental results.

| Parameter | Theoretical Result (m/s$^2$) | Experimental Result (m/s$^2$) | Error |
|---|---|---|---|
| Acceleration | 65.43 | 50.36 | 23.03% |

## 6. Conclusions

The dynamics optimization research and dynamics accuracy and reliability analysis of a complex multi-link mechanism containing clearances are researched in this paper.

(1) The dynamics optimization model and dynamics accuracy and reliability model of a mechanism are built, considering the clearances.

(2) The effects of two different optimization objective functions on the dynamics optimization of a mechanism with clearances are compared and analyzed, and the effects on dynamics responses and the nonlinear characteristics of the mechanism before and after optimization are analyzed. It is found that optimization effectively improves the dynamics performance of the mechanism considering clearances; it weakens the mechanism's chaos phenomena and improves the mechanism's stability.

(3) The dynamics accuracy and reliability of the mechanism containing clearances before and after optimization is studied. It is found that the optimization effectively improves the reliability of the mechanism containing clearances.

(4) A test platform was built to study the dynamics of the mechanism with clearances. The theoretical result and the experimental result are basically consistent in terms of trends, but there is a certain difference in value. The error rate between the theoretical result and the experimental result is 23.03%, which basically verifies the correctness of the theoretical results.

**Author Contributions:** Writing original draft, S.J.; Formal analysis, Y.L.; Investigation, J.L.; Methodology, L.X.; Funding acquisition, S.Z. All authors have read and agreed to the published version of the manuscript.

**Funding:** This research is supported by the National Natural Science Foundation of China (grant no. 61703243).

**Institutional Review Board Statement:** Not applicable.

**Informed Consent Statement:** Not applicable.

**Data Availability Statement:** Not applicable.

**Conflicts of Interest:** The authors declare no conflict of interest.

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
