# Peer review of "Dynamics Optimization Research and Dynamics Accuracy and Reliability Analysis of a Multi-Link Mechanism with Clearances"

_machines, doi:10.3390/machines10080698_

Round 1
Reviewer 1 Report
General comments
The paper deals with the optimization of a 2 DOF nine bars mechanism containing clearances in some joints. It is mainly based on simulation with a short experimental comparison. It introduces two optimization objectives function and shows how the latter influences the optimization results.
The paper is globally correctly written. The objective are stated. The methodology is explained.
The state of the art covers a large number of publication in the field of mechanism optimization, especially multi-link mechanism. It should be completed regarding the modelling aspects, which are a key element for such a study. Consider for instance following papers:
- Acary V. Projected event-capturing time-stepping schemes for nonsmooth
mechanical systems with unilateral contact and Coulomb’s friction, Comput. Methods Appl. Mech. Engrg. 256 (2013) 224–250
- Bruls O. et al, Simultaneous enforcement of constraints at position and velocity levels in the nonsmooth generalized-α scheme, Comput. Methods Appl. Mech. Engrg. 281 (2014) 131–161
The global methodology is well explained, based on modelling of the device, accounting for clearance, and optimization of the mass and inertia parameters.
The way clearance is modeled is quite clear but should be precised. More details should be given on how impacts and the transition between contact and non-contact phase are handled. This is a key element for such simulations. Furthermore, the paper does not gives details about the time discretization scheme and numerical integration method. This is not really the topic of the paper but may be particularly critical due to the difficulty of dealing with contact and impact properly. This may disturb the numerical procedure by introducing spurious oscillation in the results, which may in turn distort the optimization process (see above mentioned reference for details).
The presented results show that the proposed optimization procedure enables to tune the system performance taking clearance into account. The practical impact of such optimization for industrial application is difficult to assess, the paper being quite theoretical.
The experimental benchmark is interesting to validate the methodology and the results. Nevertheless, the impact of experiments on the paper is quite limited and should be strengthened considering additional results, for instance to illustrate the random behaviour at the crank level or the impact of the mass parameter optimization.
Adding a general figure of the optimized mechanism illustrating how it works and emphasizing the 2 dof of the system may help to understand how it works. In particular, it is not clear why there are 2 actuators while only one output is considered (slider motion).
English is globally correct and the paper reads well, except for the 2nd part of the paper. Some typo and sentences should be double checked, especially in 2nd part of the paper.
Specific comments
Page 3, line 131: Check sentence starting at line 131. It is not very consistent with the previous one.
Page 4, in equation at line 175, it should be the square root of the scalar product of the e vector (notation with the transpose of its component is not consistent)
Page 5, line 183, Eq. 3, delta=0, is it "continuous" contact state rather than "continue"?
Page 5, at line 184, precise what is L-N model
Page 6, line 210 "rod 2" rather than "rods 2"
Page 10, why should gravity be considered in the equations? According to the picture illustrating the experiment, the mechanism on the table is horizontal. So the gravity must be set to 0 in the simulation. The paper does not precise if the equations were adapted to take that aspect into account.
Page 12, Table 3 caption: precise which rod are considered flexible
Page 12, line 337, "exagerate" rather than "exaggerate"
Page 14, line 373, The "To sum up" sentence is not clear and seems in contradiction with the previous ones (optimization 2 has more benefits than optimization 1)
Page 15, in Fig. 5, the figure and subfigure captions must be more precise. Especially for subfigure a, b and c: precise to which point corresponds position, velocity and acceleration (is it for the slider). Check caption of subfigure e.
Page 20, in fig 15, use same scale on vertical axis for the two plots (same axis limits) to make the comparison between curves easier.
Reviewer 2 Report
I like objective and material of this paper.
My remarks:
1. Define optimization parameters clearly, criteria and limitations revealed OK. What parameters are changed remain unclear.
2. Please, define research task in implicit way, for instance, at the end of introduction or as separate subchapter, because now it is unclear, how your claims on optimization efficiency fullfiled in results.
3. Please, improve your model in fig.3 - it is hard to understand essential components and key geometry as well as optimized parameters.
4. Comparison of simulation and experimental measurements begs for explanation and comparison graph or table; this is also need some place in conclusions,
Round 2
Reviewer 1 Report
Thanks for adding additional figure and precision related to comment of the first version. Paper is now clearer and I just have a few remarks listed below.
* page 4, line 175 >> Check tying error: "ande"
* page 12, line 416. Thanks for adding information about numerical treatment of the dynamics equations. Nevertheless, the given information seems contradictory: the sentecne states that Runge-Kutta is used, through the odes15s function in Matlab. According to Matlab documentation, ode15s is not based on Runge-Kutta (ode45 is). Anyway, ode15s is adapted for stiff problem, which may be the case in the present case due to the contact condition. Please clarify which numerical integration method is used by using ode15s.
* page 16-17, figure 6, subfigure a, b, c, please precise that is position, velocity and acceleration of the slider.
